# Patient information needs for transparent and trustworthy cardiovascular artificial intelligence: A qualitative study

**Austin M. Stroud**[1]*, **Sarah A. Minteer**[2,3], **Xuan Zhu**[3], **Jennifer L. Ridgeway**[3,4], **Jennifer E. Miller**[5], **Barbara A. Barry**[3,4]

1 Biomedical Ethics Program, Mayo Clinic, Rochester, Minnesota, United States of America, 2 Physical Medicine and Rehabilitation Research, Mayo Clinic, Rochester, Minnesota, United States of America, 3 Robert D. and Patricia E. Kern Center for the Science of Health Care Delivery, Mayo Clinic, Rochester, Minnesota, United States of America, 4 Division of Health Care Delivery Research, Mayo Clinic, Rochester, Minnesota, United States of America, 5 Section of General Internal Medicine, Department of Internal Medicine, Yale School of Medicine, New Haven, Connecticut, United States of America

* stroud.austin@mayo.edu

## Abstract

As health systems incorporate artificial intelligence (AI) into various aspects of patient care, there is growing interest in understanding how to ensure transparent and trustworthy implementation. However, little attention has been given to what information patients need about these technologies to promote transparency of their use. We conducted three asynchronous online focus groups with 42 patients across the United States discussing perspectives on their information needs for trust and uptake of AI, focusing on its use in cardiovascular care. Data were analyzed using a rapid content analysis approach. Our results suggest that patients have a set of core information needs, including specific information factors pertaining to the AI tool, oversight, and healthcare experience, that are relevant to calibrating trust as well as perspectives concerning information delivery, disclosure, consent, and physician AI use. Identifying patient information needs is a critical starting point for calibrating trust in healthcare AI systems and designing strategies for information delivery. These findings highlight the importance of patient-centered engagement when developing AI model documentation and communicating and provisioning information about these technologies in clinical encounters.

## Author summary

As health systems start using artificial intelligence (AI) in patient care, it is important to ensure these technologies are transparent and trustworthy. Some ways to achieve this are by creating detailed documentation about AI systems and discussing the use of AI during clinic visits. It is also crucial to understand what information patients need to feel comfortable and trust these technologies. To explore this, we conducted online focus groups with 42 patients across the United States, focusing on AI in cardiovascular care. We found that patients need specific information about an AI model, such as how it is overseen and how it will affect their health care. Patients indicated some of this

**Data availability statement:** The data and materials that support the findings of this study are openly available on the Open Science Framework (OSF) at https://doi.org/10.17605/OSF.IO/EWF2Y.

**Funding:** This work was supported by the U.S. Food and Drug Administration (FDA) of the U.S. Department of Health and Human Services (HHS), award number U01FD005938, totaling $712,431 with 100 percent funded by FDA/HHS. The FDA/HHS had no role in study design, data collection and analysis, decision to publish, or preparation of the manuscript. The contents are those of the author(s) and do not necessarily represent the official views of, nor an endorsement by, FDA/HHS, or the U.S. Government.

**Competing interests:** XZ offers scientific input to research studies through a contracted services agreement between Mayo Clinic and Exact Sciences. JEM receives grant funding through Yale University from the FDA and Arnold Ventures. JEM also serves on the advisory board of GalateaBio and the board of directors for Bioethics International. BAB receives honorariums from the Mayo Clinic Accelerate program for consultations with participating companies, funding from the FDA, and research funding through a contract between Mayo Clinic and Anumana, Inc. BAB also serves on the Mayo Clinic's Biomedical Ethics and AI Advisory Council and Enterprise AI Translation Advisory Board. AMS, SAM, and JLR report no competing interests.

information to be relevant to their evaluations of trust for AI. Patients also have views on how this information should be delivered, the importance of consent, and how doctors use AI. Understanding these patient needs is essential for building trust in healthcare AI systems and helps in designing effective ways to share information and involve patients in decision-making.

## Introduction

Healthcare artificial intelligence (AI) transparency—making the processes and practices of AI systems and their underlying data understandable to humans—relies on a complex network of stakeholders, frameworks, and methods, and emerges from multiple sub-domains [1–3]. This article's primary focus is on the sub-domain of external transparency or what we shall refer to as patient-centered transparency—"the one that is addressed toward the recipients of medical care" [1]. Transparency is critical to human determinations of when to appropriately trust or distrust AI or what is termed "calibrated trust" [4,5]. Insufficient AI transparency can lead to miscalibrated trust, and may be disruptive to physician-patient relationships which are essential for proper clinical investigation and treatment [6,7].

A key element of patient-centered transparency for AI is model interpretability [1]. AI models are the algorithmic systems that process data and generate outputs. These models power AI tools, which are practical applications that perform specific tasks for users. However, AI models are often opaque without insight into how or why a particular output was derived, raising concerns even if an output proves to be measurably accurate [8,9]. In direct support of model interpretability, there is growing scholarship around approaches to AI model documentation [10–12]. These approaches provide supplemental information and evaluative criteria about a given model, tool, or data set and, in some instances, can be dynamically tailored for different use cases or revised as models are updated. Prototypes such as "AI nutrition labels" that display information similar to nutrition or prescription drug labels have been developed for technical and clinical end-users (i.e., software engineers, data scientists, and physicians). However, there is still a lack of consensus regarding the optimal approach for including and assessing information about AI models, and these efforts have largely focused on enhancing interpretability for a technical and clinical expert audience [13–16]. Knowledge gaps remain in developing AI model documentation that reflects an understanding of information factors—discrete elements of information communicated to a patient (e.g., model performance, manufacturer)—that influence patient trust.

Other key elements of patient-centered transparency are communication and information provisioning [1]. This includes practices such as disclosing the use of AI tools, obtaining informed consent, or evaluating its outputs in shared decision-making conversations [17,18]. Although integral to transparency, regulatory guidance on how clinicians and health systems should engage in practices for provisioning patient-centered information is severely lacking. Emerging conceptual scholarship in this area has begun to address this gap. Rose and Shapiro have suggested an ethical framework outlining five evaluative criteria for when healthcare AI would require patient notification and/or informed consent [19]. Other work from Hurley et al. has identified several potential goals underpinning notice and explanation for healthcare AI, examining both ethical and practical considerations [20]. While this conceptual work is essential for advancing the theoretical foundations of ethical communication and information provisioning practices, empirical research characterizing patient perspectives is necessary to ensure these practices are responsive to patients' information needs [21].

Some empirical studies have examined different aspects of patient-centered transparency in healthcare AI. Qualitative research by Robinson et al. found that patients require clear and accessible information about AI in the context of diabetes management, emphasizing the role of various training formats to support patient understanding [22]. Kostick-Quenet et al. identified various patient and epistemic considerations for trust in a prediction algorithm in qualitative research with care providers, patients, and caregivers [23]. Additionally, a survey of South Korean patients conducted by Park identified patient preferences for disclosure of AI as part of informed consent practices in a radiology context [24]. Though empirical work in this area is nascent, these early studies have provided valuable insights into patient information needs and consent preferences, highlighting the importance of engaging patients in developing healthcare AI transparency approaches.

The present study advances prior theoretical and empirical work in patient-centered AI transparency, focusing on the elements of model interpretability and communication and information provisioning. For model interpretability, the study aimed to identify patients' core information needs for AI model documentation by ascertaining discrete information factors, particularly those that influence patient trust, along with preferences for information delivery. For communication and information provisioning, the study aimed to characterize patient perspectives and preferences on practices related to disclosure, informed consent, shared decision-making, and physician AI use. To elicit patient perspectives, we used clinical vignettes depicting AI tools in the diagnosis, treatment, and/or monitoring of cardiovascular conditions along with additional probes about AI transparency. Our overall objective is to provide a more robust evidentiary foundation for patient-centered transparency approaches.

## Methods

### Study design

This study is part of a larger multi-method investigation of clinician and patient information needs and ethical considerations for adoption of AI in healthcare [25]. We followed a social constructivist framework, which presumes that knowledge and meaning are co-constructed through interaction and experience [26]. In alignment with our epistemological assumptions, we conducted asynchronous online focus groups to elicit patient perspectives [27]. In this variation of focus groups, participants engage in text-based discussions over an extended period rather than a live setting. This method supported both participant and moderator interactions, interpretations, and reflections in the course of generating data [28]. There were also pragmatic reasons for selecting an asynchronous online variation. The format afforded patients greater flexibility and convenience to participate in the study and gave an extensive period to reflect on and contribute their responses compared to traditional focus groups [29]. Additionally, literature suggests that text-based focus groups generate similar thematic content and achieve similar data quality when compared to live variations, despite the overall amount of content being less [30].

Focus groups generated both qualitative and quantitative data through open-ended prompts and multiple-choice survey questions, respectively. Survey questions were included to support methodological triangulation, as well as to provide participants a structured starting point for some prompts [31]. This article focuses on the study's qualitative findings, but also reports the survey data as a complement to the results. Our reporting follows guidance from the Standards for Reporting Qualitative Research (SRQR; see S1 Table) [32].

**Researcher characteristics.** The study team consisted of a multi-disciplinary group of researchers with expertise in biomedical ethics, health services research, implementation science, health communication, and human-computer interaction. Members of the team had

experience in both qualitative and quantitative research methods. The team's experiences conducting previous patient-centered research informed their perspectives and approach to the present study. While the study was conducted with a commitment to academic rigor, including reflective debriefs to assess focus group moderation practices and data quality, the researchers acknowledge that their backgrounds and experiences may have shaped interpretations of the data and findings.

## Ethics

Review and approval for this study and all procedures were obtained from the Mayo Clinic Institutional Review Board (IRB; protocol # 21-012302). Consent was obtained from participants verbally prior to each focus group in accordance with IRB guidance.

## Sampling and recruitment

Potential research participants were sampled via ResearchMatch, a national health volunteer registry with over 140,000 subjects who have agreed to be contacted about research opportunities [33]. Several academic institutions created this database; it is supported by the US National Institutes of Health as part of the Clinical Translational Science Award program.

We contacted participants via the ResearchMatch platform with a message containing a brief description of our study, eligibility criteria, and voting buttons that let subjects affirm or deny their interest. ResearchMatch provided contact information (e.g., name, email address) for those who responded affirmatively to the message. These eligible participants were added to our recruitment list. Participants were considered eligible if they were at least 18 years of age, English-literate, and had a primary care or cardiology clinical visit within the past three years. We oversampled from American Indian, Asian, Black/African-American, Native Hawaiian/Pacific Islander, Hispanic/Latino, and Multi-racial populations to increase their representation in our sample, as these groups have been underrepresented in US medical research [34,35].

## Data collection

**Clinical vignettes.** Four clinical vignettes depicting AI tools in cardiovascular care were used as prompts for discussion alongside several questions. Vignettes were included as they are known to support participant engagement with a topic, particularly when a participant may have limited familiarity with the subject [36]. Tools depicted included 1) a smartwatch that monitored electrocardiogram readings, 2) a stethoscope that classified heart murmurs, 3) a monitoring tool that assessed a patient's condition after a heart attack, and 4) an inpatient monitoring tool that assessed post-surgical sepsis risk [37–40]. These specific scenarios were selected as they sufficiently represented different attributes of healthcare AI tools (i.e., algorithm function, user, medical condition severity). For instance, the smartwatch scenario depicted a patient end-user compared to the stethoscope case featuring a clinician end-user. Vignettes were drafted by members of the study team with experience developing clinical case scenarios for qualitative research and education (AMS, BAB, XZ). The team examined US Food and Drug Administration-reviewed cardiovascular AI tools and academic literature to inform vignette development [41]. Draft vignettes were subsequently reviewed by subject matter experts for clinical accuracy and patient-centered communication and were finalized based on expert feedback and design recommendations [42]. An example vignette is available in the supporting information section (see S1 Appendix).

**Asynchronous online focus groups.** Three asynchronous online focus groups were conducted. Each focus group took place over two days on FocusGroupIt (focusgroupit.com).

Focus groups were limited in size to up to 15 participants based on best practices that balances the depth and richness of discussion and participant and moderator burden [27,43]. Questions were pre-loaded into the FocusGroupIt platform by a member of the research team (AMS). The first set of questions were posted at midnight, and the second set of questions were posted 24 hours later. This staggered approach was employed to limit participant cognitive load as well as to provide time for participants to reflect on their and other participants' responses between days. Participants responded to both free-response and multiple-choice questions to enhance the breadth of collected data. Multiple-choice questions included additional space for participants to supplement their answers with free-text comments.

Participants were encouraged to log in on two consecutive days to answer the posted questions and comment on other participants' posted responses. All participants were given an anonymous account name by FocusGroupIt when they logged in (i.e., Participant 1, Participant 2). On both days, four members of the research team (AMS, BAB, SAM, and XZ) moderated participants' posts throughout the day and posted follow-up questions and responses. After each focus group, moderators documented their personal reflections on moderation practices, data quality, and thematic saturation as part of a shared field note. Moderators subsequently met to debrief on these reflections, and insights from these debriefing sessions guided moderation practices for subsequent focus groups [44].

After participants responded to all questions, they were redirected to a separate survey hosted in REDCap where they could provide their contact information to receive $25 in remuneration [45,46]. Personal information entered in REDCap could not be linked to participants' focus group responses. Participants who had agreed to participate in the focus group but had not entered information into REDCap at the end of day two, received an email reminder that the final focus group section provided a link for entering their information for remuneration. We left focus groups open for three days in case some participants needed extra time to complete their responses. Moderators did not post any follow-up prompts after the second day.

## Data analysis

All free-text responses from each focus group were exported verbatim as transcripts. Focus group transcripts were analyzed using a rapid analysis approach employing a combination of deductive and inductive methods. This enabled the study team to efficiently code the primary data and use findings to inform subsequent study methods while also maintaining methodological rigor [47].

Three study team members conducted directed content analysis (AMS, SAM, and XZ) following a deductive approach [48]. Focus group questions were grouped into thematic headings, and matrices were created for each heading. These matrices included spaces to record both participant quotations and analyst summary comments. Analysts then coded exemplar participant quotations into these matrices. One primary analyst coded transcripts and provided initial comments, while a secondary analyst independently reviewed the matrices and transcripts to supplement additional quotations and comments. The primary and secondary analysts met to resolve any discrepancies and reach consensus. Once completed, the matrices were collapsed across focus groups and synthetic memoranda were generated that identified major themes. The full study team discussed these major themes and key findings and arrived at consensus.

A thematic map illustrating patient information needs was generated following an inductive approach. Responses to questions concerning information factors (i.e., *what would you want to know if AI was being used in your healthcare?*) and trust factors (i.e., *what would you need to trust the AI results or recommendations?)* were reviewed and categorized into thematic clusters to reveal underlying patterns and relationships. Information factors were then visually

mapped according to thematic relationships among the responses. Multiple choice demographic and attitudinal survey question responses were aggregated, and descriptive statistics were generated using R (version 4.2.1).

## Results

Eighty-five patients were contacted via email with an invitation to participate in the study. Of those who expressed interest, 43 participants were enrolled in focus groups between October and November of 2022. One participant was excluded from analysis for providing consistently off-topic responses that did not address any of the focus group discussion prompts. This resulted in a final sample of N = 42 participants across three focus groups. Demographic characteristics of study participants are presented in Table 1.

We report on main and sub-themes related to 1) information about the AI tool, oversight, and impact on care as well as trust factors, 2) preference for model documentation and information delivery via labels and other mechanisms, 3) importance of disclosure of AI use when critical to provider decision making or care provision, 4) patient perceptions of consent to use AI, 5) patient preferences for deciding whether AI is used in their care, and 6) favorable perceptions of physicians using AI. In addition to these findings, we present descriptive statistics for responses to attitudinal questions concerning AI transparency detailed in Table 2. While our findings are grounded in clinical vignettes describing AI in cardiovascular care, it is worth noting that patients often generalized their responses to broad uses of AI in their care in addition to responding to specific vignette-based questions.

### Information about the AI tool, oversight, and impact on care experience influence trust

Participants expressed information needs that generally fell into one of several major and sub-themes including information about the tool, oversight mechanisms, and impact on care as illustrated in a thematic map in Fig 1. Additional quotations highlighting each information factor are included in the supporting information section (see S2 Table).

**Information about the AI tool.** Participants wanted to know the general characteristics of AI tools used in their care such as its name, function, and purpose. Participants also wanted to know who manufactured the tool, which influenced their trust of a given system due to perceptions of reliability.

> *I would want to know the company that produced the software in the smart watch and the company that produced the smart watch. Their reliability tells more about the accuracy of the software. (FG3)*

In addition to these general characteristics of the AI tool, patients also sought more evidence-oriented information such as the training data used, past performance, and its generalizability. Discussions of AI performance by patients concentrated on tool accuracy, effectiveness, and reliability. These factors impacted perceptions of trust and effectiveness and were relevant to concerns about algorithmic bias and errors.

> *I would want to know what studies were done using the particular AI being used in my healthcare - and how reliable, effective, and what outcomes/impact on care it had. (FG3)*

> *In order to trust the results, I guess I'd want to know that people like me have been represented in the tests, clinical trials, etc. (FG2)*

**Table 1. Demographic characteristics of 42 patients who participated in focus groups.**

|  | N (%) |
|---|---|
| **Age** | |
| 18–24 | 3 (7.1) |
| 25–34 | 11 (26.2) |
| 35–44 | 10 (23.8) |
| 45–54 | 7 (16.7) |
| 55–64 | 6 (14.3) |
| 65 and above | 5 (11.9) |
| **Gender** | |
| Man | 20 (47.6) |
| Woman | 22 (52.4) |
| **Race/Ethnicity** | |
| American Indian or Alaskan Native | 3 (7.1) |
| Black or African American | 15 (35.7) |
| Hispanic/Latinx | 1 (2.4) |
| Asian | 6 (14.3) |
| Multi-race | 2 (4.8) |
| Non-Hispanic White | 15 (35.7) |
| **Education level** | |
| High school graduate or GED completed | 2 (4.8) |
| Completed a vocational, trade, or business school program | 2 (4.8) |
| Associate's degree (e.g., AA, AS) | 4 (9.5) |
| Bachelor's degree (e.g., BA, BS, AB) | 15 (35.7) |
| Master's degree (e.g., MA, MS, MSW, MBA) | 15 (35.7) |
| Doctorate or professional degree (e.g., PhD, MD) | 4 (9.5) |
| **Familiarity with AI** | |
| Very familiar | 2 (4.8) |
| Familiar | 24 (57.1) |
| Neutral | 11 (26.2) |
| Unfamiliar | 4 (9.5) |
| Very Unfamiliar | 1 (2.4) |
| **Marital status** | |
| Married | 15 (35.7) |
| Never married | 14 (33.3) |
| Separated, Divorced, or Widowed | 13 (31) |
| **Perceived financial status** | |
| Living comfortably on present income | 17 (40.5) |
| Getting by on present income | 21 (50) |
| Finding it difficult on present income | 0 (0) |
| Finding it very difficult on present income | 3 (7.1) |
| Prefer not to answer | 1 (2.4) |

**Table 2. Descriptive statistics of participant responses to attitudinal questions regarding healthcare AI transparency.**

| | N (%) |
|---|---|
| **Importance of being informed by care team about every AI use in your care** | |
| Very important/Important | 28 (66.7) |
| Neutral/Unimportant/Very unimportant | 14 (33.3) |
| **Importance of participating in decision-making regarding use of AI in your care** | |
| Very important/Important | 28 (66.7) |
| Neutral/Unimportant/Very unimportant | 14 (33.3) |
| **Importance of getting information about how AI is used in your care** | |
| Very important/Important | 30 (71.4) |
| Neutral/Unimportant/Very unimportant | 12 (28.6) |
| **Do you think labels for AI software are needed?** | |
| Yes | 33 (78.6) |
| No | 9 (21.4) |
| **Importance of provider explaining how they made a decision about your care based on AI result** | |
| Very important/Important | 31 (73.8) |
| Neutral/Unimportant/Very unimportant | 11 (26.2) |

Participants also wanted to be informed about the tool's limitations and potential risks.

*I would want to know about any limitations needed while using it, which the provider would usually review with me anyway. Trust results if verified as correct by provider. (FG1)*

Overall, these information factors related to the research, design, development, and validation of a given AI tool along with basic information that might guide use. These were often descriptive elements about the tool itself or secondary evaluations of its capabilities that could inform patient understanding.

**Information about oversight.** Patients wanted to know what oversight procedures were in place for any AI tools used in their care which was also a factor influencing trust. Effective oversight was described as a distributed and layered process undertaken by regulatory bodies, care teams, and even other patients.

*I would imagine a board with medical providers and patient representatives setting rules and procedures for all kinds of situations and monitoring compliance. (FG1)*

*I would feel comfortable, knowing fully well it's been approved by FDA and other relevant agencies governing it's [sic] usage. (FG3)*

Additionally, a recurring theme was patients' preference for their care teams to review and confirm the insights from AI tools used in their care, leading to greater trust.

*Trust in the AI will come when my healthcare team reviews the data collected and agrees with the AI generated recommendations. (FG1)*

Lastly, participants felt strongly that there should be transparent and reliable oversight mechanisms in place for the use and storage of their data. They wanted to know what data stewardship practices were in place with particular concern for whether their data might be shared or sold, which influenced their trust in these systems.

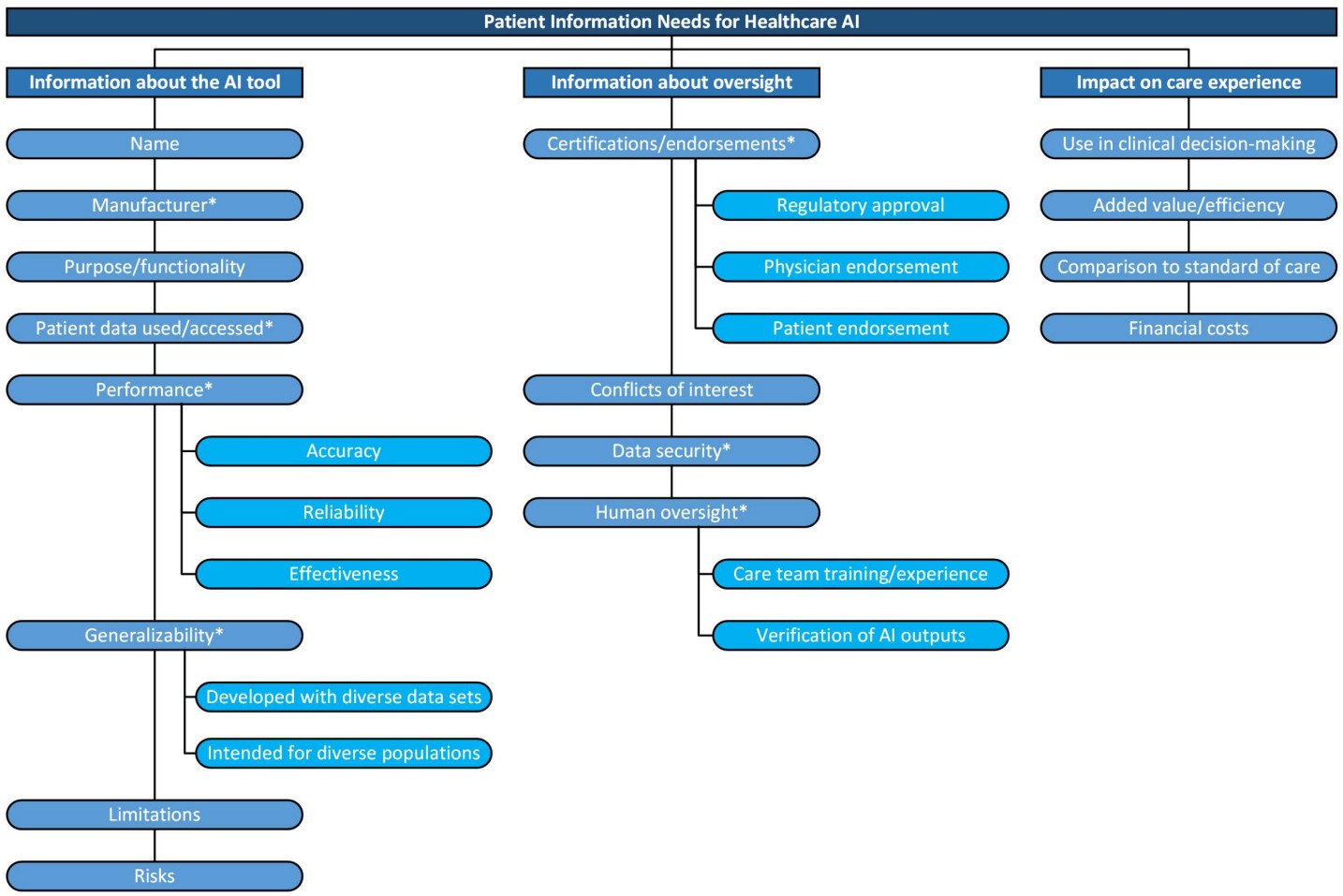

**Fig 1. Thematic map of patient information needs for healthcare AI.** High-level categories of information needs identified from inductive analysis include information about the AI tool, information about oversight, and the impact on the care experience. Information factors are enumerated for each category. Asterisks indicate factors that mediate trust.

*I see this as a regulatory concern. I would think that AI presents a big logistical challenge with regard to sharing patient data culled from medical records. Privacy and confidentiality need to somehow be maintained. (FG1)*

These information factors related to processes for ensuring safe and effective use of AI tools and patient data. A differentiating aspect of these factors was their connection to organizations or stakeholders whose responsibility is oversight. Patient trust was less so connected to information about the AI and instead relied upon a fundamental trust in the organizations or stakeholders responsible for ensuring safety and efficacy.

**Impact on care experience.** Participants wanted to know precisely how AI tools would be used in their care such as how AI would factor into clinical decision making, the added value AI provides, and how care involving AI compared to standard of care procedures.

*I would appreciate transparent and clear communication about the role and impact of utilizing AI in that healthcare service setting. (FG3)*

*I would want to know how it is done and comparable to an in-office test, something that was told to me before I actually got the warning to trust that it is accurate. (FG2)*

Participants also desired information on any financial costs associated with the use of AI in their care.

*I would want to know how much this [AI] is costing me. Especially if this is something you would usually have as a follow up with a medical professional. (FG2)*

These factors were differentiated in that they relied on the specific clinical context in which AI might be used by a patient's care team. As a result, they were the most dynamic and could vary greatly depending on the specific AI tool and circumstances for its use. Nonetheless, patients expressed a desire to know how AI might shape their care, ranging from specific care decisions to downstream cost considerations.

## Preference for model documentation and information delivery via labels and other mechanisms

Most participants were receptive to the idea of labeling AI tools. This was reinforced by the multiple-choice question data, with 78.6% (n = 33) participants responding that labels would be necessary for the use of AI tools in their care. Patients also emphasized that labels should be designed with an emphasis on interpretability for patients.

*Labels for AI software are a great idea, in principle. They need to be understandable to the layman. (FG1)*

However, some participants challenged the idea of an informational label suggesting that they might contribute to misunderstanding. Labels could potentially create confusion or fail to adequately communicate complex information about AI models.

*I think the struggle with this is the communication of what these labels mean. It is not common knowledge of everything that is described on nutrition labels, and I think it would be similar and could lead to confusion in the health care setting further. (FG2)*

*Labels are good, but might oversimplify the precision and huge amount of data, processing and nuances that go into any AI technology. (FG3)*

Patients also suggested other mechanisms for providing information on AI tools, including leveraging consent forms, informational videos, visit summaries, or discussions with their care team.

*I think that any information that could be included on a label is best contained in a consent form signed with your doctor for treatment of a specific condition. Since different AI software would be used for different scenarios, the label would vary accordingly. It needs to be linked to the specific condition it is being used to treat. (FG1)*

In addition to specifying *how* they wanted information delivered, participants also commented on *who* they would like to receive information from. Participants often stated wanting to be informed by their treating physician or other member of the care team.

*Practically I would love the information about this AI from an experienced practitioner. I would prefer the information be explained to me by an expert either directly or through emails. (FG1)*

Patients also discussed the appropriate timing for receiving information, with many participants wanting to be informed about the AI tool prior to or during its use for tools that are higher risk or are addressing more immediate concerns. Participants also indicated information about the AI tool used could be included in post-visit documentation such as an after-visit summary.

*I would want to be notified on the spot directly from my Doctor, I would like the info presented to me both on paper in an after visit summary. I would also like it to be verbalized to me by that same Doctor. (FG2)*

## Importance of disclosure of AI use when critical to provider decision making or care provision

On the multiple-choice survey questions, 66.7% (n = 28) participants indicated that it was very important or important to be informed by their care teams about every use of AI in their care, and 71.4% (n = 30) participants responded that it was very important or important to receive information about how AI is used in their care. Focus group responses also indicated, most patients wanted to be informed about AI use to understand the impact on their care and how it might factor into a care team's decision.

*AI is a tool that can be used by the team and I would like to know how it is used. Since AI is a relatively new technology it is important that the patient is informed in order to build trust in AI. (FG1)*

Some patients noted that the context of AI use influenced their perceived importance of its disclosure; patients wanted AI to be disclosed if it played a critical role in their care (e.g., surgical procedures, diagnoses, monitoring), but were ok with not being informed about less consequential uses (e.g., scheduling, notifications).

*In small things like appointment scheduling, I think it is obvious it is AI and I do not need to be informed for it. In bigger things like device monitoring I would probably like to know. I think knowing for me is more so just because if something goes wrong, I will know who to contact or what to do. (FG3)*

Participants were mixed on being alerted to AI findings and, if so, how. Some welcomed being provided with all data gathered by AI, while others only wanted to be alerted by their doctor if AI detected something serious, warranting treatment. Some worried that notifications about minor findings could cause undue alarm.

*I think that a lot of caution is needed in presenting AI analysis results. They could cause a lot of unnecessary stress to individuals involved. I also think this need[s] ethical considerations. What if an AI analysis predicts a non-curable, terminal illness? What are the ethical considerations for that kind of analysis and information. (FG1)*

## Variability in patient perceptions of consent to use AI

Participants expressed mixed opinions regarding the need to provide consent for AI to analyze their data. Some thought their data was likely already being used and that consent was already covered under agreement to be treated.

*I could not care less. I almost have some expectation that it [AI] will be utilized without my knowledge or consent. I don't see the need to make this any different than a doctor using any computer system to assist them. (FG2)*

Others valued consent but did not see the need for it to be provided each time AI was used. Some patients thought a broad consent would suffice; others wanted to provide consent for particular uses. For instance, participants were more concerned about consenting when their personal information might be shared with third-parties and less concerned when used by their care team for treatment.

*I would feel uneasy about this unless it was disclosed and I consented to the continuous monitoring. Once consented, similar to the waiver for all healthcare data collected at [Clinic] to be open for research purposes, I found this honest and innovative that my data may help others in the future. (FG2)*

Finally, some participants wanted to provide consent every time AI was analyzing their data.

*I would have to consent to any of this. It sounds like overkill. I would be concerned about the security of the information. (FG1)*

## Variability in patients' preferences for deciding whether AI is used in their care

On survey responses, 66.7% (n = 28) participants indicated that it was very important or important to be involved in decision-making regarding the use of AI in their care. Furthermore, 73.8% (n = 31) responded that it was important to have a provider explain how a decision was made based on an AI result. Patients expressed varied opinions on needing to be involved in decision-making concerning healthcare AI. Many felt they should be involved in the decision to use AI just as they would want to actively make decisions about other aspects of their treatment.

*My decision must be required before I can allow a machine work on me even if it's 100% accurate in its dealings and more accurate than humans. (FG3)*

Other participants envisioned leaning on the expertise of their care team, but still maintaining authority in deciding whether AI would be used.

*If the provider is supposed to be an expert, I expect them to give me their opinion about what's best. Of course, as a patient, I would always have to [the] right to yay or nay its use. That said, I always anticipate being a partner in my healthcare decisions with any providers. (FG2)*

Some patients were comfortable deferring to their care team's judgment, suggesting that they would have more expertise to make an informed decision. However, informing patients about the use of AI was still desired.

*I'm not sure I'm qualified enough to make an informed decision about its use. I would like to know if and how it's being used but I would leave the decision up to the care team. (FG1)*

## Mostly favorable perceptions of physicians using AI

Participants generally expressed that their opinion of their provider would not change if their provider used AI in their practice. Furthermore, others trusted their physician's judgment on when AI might be a helpful tool based on the physician's clinical knowledge.

*It [AI] would not impact my trust for the experienced physician at all because ultimately, I trust that physicians use tools and experience to come up with the diagnosis. The AI is there as a technologically enhanced tool. (FG1)*

Many viewed the use of AI as a positive sign that their physician was keeping up to date with new technology and using it in conjunction with their clinical expertise to provide care and make recommendations.

*It wouldn't change my opinion in a bad way but might in a good way. For example, I love to read and am often surprised that I might have read something that a clinician has not. It makes me think that they don't try hard enough to keep up with new research. If I discovered that a provider was using AI software, I think I'd think more highly of them, depending on the purpose of the software. (FG2)*

The extent to which participants had cultivated relationships with their physicians over significant periods of time contributed to this level of trust, irrespective of the tools used.

*It would not change my opinion of my healthcare provider. How I feel about him/her is based on personal interaction, knowledge of my medical history, and the trust that has been built up over the years. (FG1)*

In circumstances where a physician disagreed with an AI recommendation, most participants would not view the physician negatively. Some patients would even view their physician more positively for not deferring to the AI recommendation.

*I would believe that the provider is thinking critically about the results of the AI's interpretation of the data it was given, and as long as the explanation and rationale is given plausibly and logically, I would be reassured by the disagreement. If it kept happening though repeatedly in a pattern, it would concern me that something systemically, on either (or both) [the] part of the prescriber or AI, is wrong. (FG3)*

However, some participants would want reassurance if their physician disagreed with the AI's recommendation and might seek out a second opinion from another physician.

*I usually do some research before even seeing a provider so I have a notion of what they'll say and even prescribe. If a doctor disagrees with an AI's result but still aligns with what I've found on my own, that would be helpful for me but I think I'd still have to seek a second opinion and let them know why I was doing that. (FG2)*

## Discussion

The present study is among the first to systematically identify information needs and factors that facilitate trust in AI from the perspective of patients. Our findings identify discrete information factors that form the basis for creating patient-centered, empirically supported, and

interpretable AI model documentation. Furthermore, our results add to the growing body of work in this area by expanding, affirming, and characterizing patient information needs and trust factors identified from other specialty areas. Our study also provides exploratory findings on patient preferences for communication and information provisioning of AI models and tools used in their health care. Results indicating varied perspectives on disclosure, consent, and shared decision-making may support health systems and clinicians on how to respond to these preferences when deploying AI. Finally, our work supports past research demonstrating transparency is important for calibrating trust in the use of AI systems in healthcare and physicians who choose to use AI tools [23,49].

Certain information factors raised in our study including performance, risks, endorsements, and human oversight support findings from prior empirical work examining patient perspectives on devices in diabetes management and predictive algorithms in left ventricular assist device therapy [22,23]. This might indicate the existence of general patient information needs independent of the specific AI model or clinical specialty. Beyond supporting prior findings, our study expands the breadth of information factors that may be relevant to patients, including additional considerations related to descriptive information about AI tools, aspects related to oversight, and specifics regarding how AI will impact patient care. Prior work also suggests that factors related to model performance/accuracy and endorsements mediate patient trust. However, our study presents additional information factors that may be relevant to trust, including the specific manufacturers of AI tools, practices related to accessing and securing patient data, and design considerations for including diverse patient populations. These additional information factors and trust considerations contribute to a more robust understanding of patient-centered model interpretability for future research and transparency initiatives.

While there is consensus around the need for greater model interpretability, the best way to deliver key information remains less clear [2]. Many, both internal and external to the healthcare sector, have proposed AI model documentation in the form of AI labeling and sharing of associated data sets [11,13,16,50,51]. Dynamic AI labeling, in particular, has been argued to be a critical part of supporting overall regulatory approaches for healthcare AI [52]. Our results show that while patients tend to think labeling could improve transparency, labels may not always be the best or most comprehensive approach for communicating this information, challenging the applicability of this approach with patients. Future research focused on labeling as a type of AI model documentation may consider experimenting with different communication platforms and formats for information delivery. Labels instead might serve as an entry point to a broader ecosystem of information that patients can engage with according to their needs and preferences.

Patients have concerns about labels being confusing and inaccessible and therefore may prefer alternative ways of delivering this information (e.g., consent forms). This finding highlights the importance of incorporating diverse patient perspectives when designing strategies to disseminate information about AI tools and tailoring these strategies according to patient needs, preferences, and values. For instance, progressive disclosure could be used as an approach to allow patients to decide the depth of information they would like to review about AI tools used in their care, and materials could then be created accordingly (e.g., label with brief facts, patient handouts with more in depth data and information) [53]. Visual and interactive aspects of model documentation could also be considered in support of patient needs, and further highlights the relevance of user interactions with AI systems when designing for transparency [12,54]. These may complement other approaches such as in-person trainings, informational videos, and pamphlets, raised as potential methods to support patient training needs [22]. Additional research is needed to identify what mechanisms are most effective at

delivering information and calibrating trust in AI systems and the care teams that employ them, and how to tailor delivery mechanisms to accommodate diverse patient preferences and clinical scenarios.

Our findings related to communication and information provisioning suggest that transparency for patients encompasses more than information availability. Patients often express a preference for being an active decision-maker in their care and use of AI in it, albeit to varying degrees. These results provide an in-depth patient perspective to related quantitative research suggesting that patients prefer the disclosure of AI used in their health care and in some views may require informed consent [24]. The presence of healthcare AI in clinical settings may prompt additional conversations between patients and their care teams regardless of whether labels or other delivery mechanisms were used to communicate the information. While ubiquitously stated in ethical frameworks and guidelines, transparency is often complicated by the audience AI systems need to be transparent to [19,20,55–57].

Moreover, our findings suggest physicians play a pivotal role in facilitating AI transparency and mediating patient trust. Participants often appealed to their physician(s)' judgment on whether AI tools are safe and effective in their care. This is consistent with findings from other patient-centered research where patients expected their physicians to ensure safety of AI used in their health care [58]. These insights emphasize a need to further understand how physicians develop trust in AI tools and how that trust may then be shared with patients [49,59]. Furthermore, there are numerous implications for physicians in the adoption of AI tools, such as training, disclosure, informed consent, and liability that are relevant to how associated risks may be communicated to patients [17,24,60,61]. These additional responsibilities for physicians and healthcare institutions should be evaluated as regulations are considered for AI systems.

This study raises several considerations for key stakeholders in healthcare AI adoption and development. Clinicians and health systems who are integrating AI models and tools into patient care may wish to evaluate their protocols for provisioning information during disclosure and informed consent. Such evaluations might focus on what information is being communicated to patients, how exactly that information is communicated, and what bearing this information has on shared decision-making conversations. Clinicians might also critically evaluate their own communication skills and depth of knowledge when responding to patient inquiries about AI's role in their health care. AI developers might consider supplemental resources that support clinical and technical end-user communication with patients in addition to more technically focused model documentation. In the case of AI tools where patients are end-users, developers might include documentation that supports conversations with healthcare professionals. Regulatory entities might consider the specific information factors raised by patients while developing guidelines that establish transparent information standards and provisioning practices. We note that these are preliminary considerations and should not be construed as policy recommendations. While our findings speak to various actions key stakeholders might take, additional research is necessary assess the salience of various information factors in healthcare decision-making to synthesize robust policy and guidance.

Developing patient-centered transparency approaches fits into a broader range of strategies for enhancing overall transparency of these systems in healthcare, including for clinical and technical stakeholders. This manuscript focuses on the information needs of patients. This focus is particularly important given the dearth of research with patient populations, even though their personal data will likely be used for the development of healthcare AI systems, and they will be on the receiving end of clinical decisions driven by these tools [21,62]. By including patient perspectives in ongoing transparency work, there are opportunities to

support health equity and foster trust in healthcare AI tools [21,63]. In coordination with technical approaches, ethical frameworks, and regulatory guidelines, there is the ability to strengthen healthcare AI transparency for all stakeholders in accordance with their precise information needs [57].

## Limitations

Our study has several limitations. Conducting focus groups in an online asynchronous format can constrain the moderators' ability to ask follow-up and clarifying questions [27]. Some follow-up questions did not receive a response, and sometimes there was a time delay in correcting misunderstandings to probes. Participants were also limited to those who had sufficient broadband access and technological literacy to participate in an online hosted discussion [64]. Additionally, our findings might have been impacted by selection bias, limiting their generalizability. ResearchMatch helped facilitate recruitment beyond a single healthcare institution and local geographic region. However, as an online health research-oriented database, the pool of potential participants may have possessed greater health and technological literacy than that of the general populace which could have resulted in skewed data quality.

Participants were prompted to respond to uses of AI in healthcare broadly and more specifically through clinical vignettes based on four AI applications in cardiovascular care. However, there is immense variety in healthcare AI between numerous clinical specialties. Thus, further research evaluating a broader range of technologies and their use in clinical specialties is necessary to better understand the nuances of information needs as they relate to specific AI applications and care circumstances. While our qualitative methodology is well suited to explore the depth of participant perspectives by capturing their authentic, self-interpreted insights, this is only one process for understanding patient information needs. Additional research with larger study samples is needed to better understand how these factors might be prioritized by patients for potential transparency solutions. Future research should further engage underrepresented populations and communities through participatory research methods to examine whether and how AI information needs differ across patient demographics, with the goal to inform AI information dissemination strategies and policies that are better tailored to varying community needs.

## Conclusion

As healthcare stakeholders grapple with the adoption of AI into clinical settings, consideration for patient information needs will be vital for calibration of trust and determinations about safe and effective use of these tools. The present study outlines several of those needs along with key patient perspectives on information delivery, informed consent, decision-making involvement, and physician AI use. While some of these factors overlap with AI model documentation outlined for technical and clinical experts, insights from patients can aid the expansion of these approaches to better support patient-centered adoption of healthcare AI.

## Supporting information

**S1 Table.  Standards for reporting qualitative research (SRQR). Adapted from [32].**
(DOCX)

**S2 Table.  Supplemental quotations for information factors.**
(DOCX)

**S1 Appendix.  Example clinical vignette.**
(DOCX)

## Author contributions

**Conceptualization:** Xuan Zhu, Jennifer L. Ridgeway, Jennifer E. Miller, Barbara A. Barry.

**Data curation:** Austin M. Stroud, Sarah A. Minteer, Xuan Zhu, Barbara A. Barry.

**Formal analysis:** Austin M. Stroud, Sarah A. Minteer, Xuan Zhu, Barbara A. Barry.

**Funding acquisition:** Jennifer E. Miller, Barbara A. Barry.

**Investigation:** Austin M. Stroud, Sarah A. Minteer, Xuan Zhu, Barbara A. Barry.

**Methodology:** Xuan Zhu, Jennifer L. Ridgeway, Barbara A. Barry.

**Supervision:** Jennifer L. Ridgeway, Barbara A. Barry.

**Writing – original draft:** Austin M. Stroud, Sarah A. Minteer, Xuan Zhu, Jennifer L. Ridgeway, Barbara A. Barry.

**Writing – review & editing:** Austin M. Stroud, Sarah A. Minteer, Xuan Zhu, Jennifer L. Ridgeway, Jennifer E. Miller, Barbara A. Barry.

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
