## [Decision Letter · Decision Letter 0]

15 Oct 2024

PDIG-D-24-00340

Patient information needs for transparent and trustworthy artificial intelligence in healthcare: A qualitative study

PLOS Digital Health

Dear Dr. Austin M. Stroud,

Thank you for submitting your manuscript to PLOS Digital Health. After careful consideration, we feel that it has merit but does not fully meet PLOS Digital Health's publication criteria as it currently stands. Therefore, we invite you to submit a revised version of the manuscript that addresses the points raised during the review process.

Please submit your revised manuscript within 60 days Dec 14 2024 11:59PM. If you will need more time than this to complete your revisions, please reply to this message or contact the journal office at digitalhealth@plos.org. Please include the following items when submitting your revised manuscript:

We look forward to receiving your revised manuscript.

Kind regards,

Hadi Ghasemi

Academic Editor

PLOS Digital Health

Hadi Ghasemi

Academic Editor

PLOS Digital Health

Journal Requirements:

1. We ask that a manuscript source file is provided at Revision. Please upload your manuscript file as a .doc, .docx, .rtf or .tex.

Additional Editor Comments (if provided):

Reviewers' comments:

Reviewer's Responses to Questions

**Comments to the Author**

1. Does this manuscript meet PLOS Digital Health’s publication criteria ? Is the manuscript technically sound, and do the data support the conclusions? The manuscript must describe methodologically and ethically rigorous research with conclusions that are appropriately drawn based on the data presented.

Reviewer #1: Yes

Reviewer #2: Partly

Reviewer #3: Yes

Reviewer #4: Yes

2. Has the statistical analysis been performed appropriately and rigorously?

Reviewer #1: N/A

Reviewer #2: Yes

Reviewer #3: Yes

Reviewer #4: Yes

3. Have the authors made all data underlying the findings in their manuscript fully available (please refer to the Data Availability Statement at the start of the manuscript PDF file)?

Reviewer #1: Yes

Reviewer #2: Yes

Reviewer #3: No

Reviewer #4: Yes

4. Is the manuscript presented in an intelligible fashion and written in standard English?

PLOS Digital Health does not copyedit accepted manuscripts, so the language in submitted articles must be clear, correct, and unambiguous. Any typographical or grammatical errors should be corrected at revision, so please note any specific errors here.

Reviewer #1: Yes

Reviewer #2: Yes

Reviewer #3: Yes

Reviewer #4: Yes

5. Review Comments to the Author

Please use the space provided to explain your answers to the questions above. You may also include additional comments for the author, including concerns about dual publication, research ethics, or publication ethics. (Please upload your review as an attachment if it exceeds 20,000 characters)

Reviewer #1: Dear Authors,

I want to thank you for the opportunity to review your manuscript entitled "Patient information needs for transparent and trustworthy artificial intelligence in healthcare: A qualitative study." Your work is a valuable contribution to the growing body of literature addressing the intersection of artificial intelligence and healthcare, offering a novel exploration of patient perspectives.

While I commend the novelty of your study, I would like to offer some suggestions for improvement that further enhance your manuscript's clarity and methodological rigor.

1. Title suggestion: given that your research is conducted within the context of cardiovascular healthcare, I recommend reflecting this specific context in the title to make it more precise and to attract a more targeted readership.

2. research Design section: the design paragraph would benefit from revision, particularly to avoid redundancy in the description of the data collection strategy. This section is better suited to discussing the epistemological framework underpinning your study, which is constructivism. Clearly articulating this constructivist perspective will help anchor the study's methodology, enhancing transparency for readers.

3. reporting in general: I recommend adhering to established guidelines for reporting qualitative research, such as the "Consolidated Criteria for Reporting Qualitative Research" (COREQ) or the Standards for Reporting Qualitative Research (SRQR). These guidelines will strengthen the rigor of your study and ensure consistency in reporting.

4. Data collection (or as I would like you to say: Data generation or data construction): I encourage you to explain why you chose asynchronous focus groups. While this innovative method is needed, a stronger justification is required to clarify why this approach was deemed suitable for your specific research questions and context. Please also consider to better explain data generation which includes quantitative and qualitative data. This methodological paper may be useful: Braun, V., Clarke, V., Boulton, E., Davey, L., & McEvoy, C. (2020). The online survey as a qualitative research tool. International Journal of Social Research Methodology, 24(6), 641–654. https://doi.org/10.1080/13645579.2020.1805550

5. Data Analysis: the data analysis approach is somewhat unclear. It is essential to specify whether your analysis followed an inductive, deductive, or combined approach and why you chose to use focus group questions as initial categories for analysis. Additionally, the interpretative contribution of the authors needs to be revised. I suggest reconsidering the findings' narrative to reflect a more interpretative stance to align with a constructivist approach. Please just simply listing themes without accompanying interpretation, as this limits the depth of your analysis.

6. Rigor: I suggest adding a section on rigor in your methodology, addressing the steps you took to ensure trustworthiness and credibility, such as triangulation, member checking, or reflexivity.

7. Method and findings Overlap: please ensure that findings are not inserted into the methods section. This conflation detracts from the clarity of the research process and can need to be clarified regarding the distinction between data collection and interpretation.

8. In your limitations, avoid the conventional statement about the lack of generalizability in qualitative research. If generalizability is discussed, I recommend framing it positively by emphasizing the idiographic nature of qualitative research, which is focused on depth and contextual understanding rather than statistical generalization.

Thank you for considering these suggestions, and I look forward to seeing the revised manuscript.

Reviewer #2: 1. Lack of Actionable Recommendations: While the findings highlight patient-centered engagement, the text does not provide specific strategies or recommendations for implementing these insights in clinical practice.

2. Complex Sentence Structure: The introduction features long and convoluted sentences, which can obscure the main points. For instance, the sentence discussing "calibrated trust" could be simplified for better understanding. Some sentences are lengthy and complex, which can hinder readability. Breaking them into shorter, more concise statements would improve clarity.

3. Repetition of Concepts: Some ideas, such as the importance of transparency for trust, are reiterated without adding new insights. Streamlining these points could improve conciseness.

4. Lack of Context for Previous Research: The introduction references various studies but does not provide enough context about their findings or relevance to the current study.

5. Potential Biases in Recruitment: The recruitment strategy mentions using ResearchMatch but does not address potential biases in participant selection or limitations of this approach. Acknowledging these issues would provide a more balanced view.

6. Recommendations: Elaborate on Vignette Selection: Offer a more thorough explanation of why the specific vignettes were chosen and how they cover different aspects of healthcare AI.

Reviewer #3: The background and relevance have been clearly established.

The study design including the recruitment criteria are adequately explained. The results and discussion are insightful and aligning with the study's objectives.

Reviewer #4: Thank you for inviting me to review this paper. This has been informative and educative for me. The manuscript generally offers new insights and new data, and it is clear that this area is under-research. I believe this paper will offer the required insights for others to follow suit. I must also add that the subject matter area (Artificial Intelligence) is not one that I am very familiar with but has limited knowledge in the area. This may affect my comment on this manuscript to an extent.

Background

That being said, I am confident that the manuscript has a very good rigor to it. I was particularly interested in questioning the methodology (being that the qualitative approach is my research strength) but found that every decision was justified scientifically. However, my concern regarding the methodology is the richness of the responses provided by the study's virtual participants. I am saying this because speaking has been established to communicate feelings, attitudes, and perspectives more than informal writing. Personally, I also believe that people tend to exhibit laziness in responding to questions through writing (as long as they are not taking examinations).

Method Section

While I agree that we can generate quantitative data in a qualitative paper, the authors still need to emphasize how or why they had to include multiple-choice questions. This, I believe was not clear in the paper.

Result Section

Please shrink the excerpts to 1.0 line spacing in place of what is currently used. It helps to clearly distinguish the excerpts and the manuscript’s texts.

In several sub-themes, you have only included one excerpt. This, I believe may be limiting for your data (particularly because your participants are across the country). What validates the quality of qualitative research is identifying patterns in the participants’ responses, which is not reflected in some components of your study result. In other words, I find it more plausible if you can have a supplementary file where more excerpts per theme and/or sub-themes are included

Limitation Section

Kindly provide a heading for the Limitation section

I agree with all the limitations called out.

Conclusion

This reads well for me

6. PLOS authors have the option to publish the peer review history of their article (what does this mean? ). If published, this will include your full peer review and any attached files.

**Do you want your identity to be public for this peer review?** For information about this choice, including consent withdrawal, please see our Privacy Policy .

Reviewer #1: Yes: Luca Ghirotto

Reviewer #2: No

Reviewer #3: No

Reviewer #4: No

---

## [Decision Letter · Decision Letter 1]

3 Mar 2025

PDIG-D-24-00340R1Patient information needs for transparent and trustworthy cardiovascular artificial intelligence: A qualitative studyPLOS Digital Health Dear Dr. Austin M. Stroud, Thank you for submitting your manuscript to PLOS Digital Health. After careful consideration, we feel that it has merit but does not fully meet PLOS Digital Health's publication criteria as it currently stands. Therefore, we invite you to submit a revised version of the manuscript that addresses the points raised during the review process. Please submit your revised manuscript within 30 days Apr 02 2025 11:59PM. If you will need more time than this to complete your revisions, please reply to this message or contact the journal office at digitalhealth@plos.org. Please include the following items when submitting your revised manuscript:* A rebuttal letter that responds to each point raised by the editor and reviewer(s). You should upload this letter as a separate file labeled 'Response to Reviewers '. This file does not need to include responses to any formatting updates and technical items listed in the 'Journal Requirements' section below.* A marked-up copy of your manuscript that highlights changes made to the original version. You should upload this as a separate file labeled 'Revised Manuscript with Track Changes '.* An unmarked version of your revised paper without tracked changes. You should upload this as a separate file labeled 'Manuscript '. If you would like to make changes to your financial disclosure, competing interests statement, or data availability statement, please make these updates within the submission form at the time of resubmission. Guidelines for resubmitting your figure files are available below the reviewer comments at the end of this letter. We look forward to receiving your revised manuscript. Kind regards, Hadi GhasemiAcademic EditorPLOS Digital Health Hadi GhasemiAcademic EditorPLOS Digital Health Leo Anthony CeliEditor-in-ChiefPLOS Digital Healthorcid.org/0000-0001-6712-6626 **Additional Editor Comments (if provided):****Reviewers' Comments:** Reviewer's Responses to Questions

**Comments to the Author**

1. If the authors have adequately addressed your comments raised in a previous round of review and you feel that this manuscript is now acceptable for publication, you may indicate that here to bypass the “Comments to the Author” section, enter your conflict of interest statement in the “Confidential to Editor” section, and submit your "Accept" recommendation.

Reviewer #1: All comments have been addressed

Reviewer #4: (No Response)

2. Does this manuscript meet PLOS Digital Health’s publication criteria ? Is the manuscript technically sound, and do the data support the conclusions? The manuscript must describe methodologically and ethically rigorous research with conclusions that are appropriately drawn based on the data presented.

Reviewer #1: Yes

Reviewer #4: Yes

3. Has the statistical analysis been performed appropriately and rigorously?

Reviewer #1: Yes

Reviewer #4: N/A

4. Have the authors made all data underlying the findings in their manuscript fully available (please refer to the Data Availability Statement at the start of the manuscript PDF file)?

Reviewer #1: Yes

Reviewer #4: No

5. Is the manuscript presented in an intelligible fashion and written in standard English?

Reviewer #1: Yes

Reviewer #4: Yes

6. Review Comments to the Author

Reviewer #1: Thanks for engaging with the comments and providing a new version of the manuscript.

Reviewer #4: None of the feedback i provided was responded to or incorporated. You can find them here below

Thank you for inviting me to review this paper. This has been informative and educative for me. The manuscript generally offers new insights and new data, and it is clear that this area is under-research. I believe this paper will offer the required insights for others to follow suit. I must also add that the subject matter area (Artificial Intelligence) is not one that I am very familiar with but has limited knowledge in the area. This may affect my comment on this manuscript to an extent.

Background

That being said, I am confident that the manuscript has a very good rigor to it. I was particularly interested in questioning the methodology (being that the qualitative approach is my research strength) but found that every decision was justified scientifically. However, my concern regarding the methodology is the richness of the responses provided by the study's virtual participants. I am saying this because speaking has been established to communicate feelings, attitudes, and perspectives more than informal writing. Personally, I also believe that people tend to exhibit laziness in responding to questions through writing (as long as they are not taking examinations).

Method Section

While I agree that we can generate quantitative data in a qualitative paper, the authors still need to emphasize how or why they had to include multiple-choice questions. This, I believe was not clear in the paper.

Result Section

Please shrink the excerpts to 1.0 line spacing in place of what is currently used. It helps to clearly distinguish the excerpts and the manuscript’s texts.

In several sub-themes, you have only included one excerpt. This, I believe may be limiting for your data (particularly because your participants are across the country). What validates the quality of qualitative research is identifying patterns in the participants’ responses, which is not reflected in some components of your study result. In other words, I find it more plausible if you can have a supplementary file where more excerpts per theme and/or sub-themes are included

Limitation Section

Kindly provide a heading for the Limitation section

I agree with all the limitations called out.

Conclusion

This reads well for me

7. PLOS authors have the option to publish the peer review history of their article (what does this mean? ). If published, this will include your full peer review and any attached files.

**Do you want your identity to be public for this peer review?** For information about this choice, including consent withdrawal, please see our Privacy Policy .

Reviewer #1: **Yes: ** Luca Ghirotto

Reviewer #4: No

---

## [Decision Letter · Decision Letter 2]

17 Mar 2025

Patient information needs for transparent and trustworthy cardiovascular artificial intelligence: A qualitative study

PDIG-D-24-00340R2

Dear Austin M. Stroud,

We are pleased to inform you that your manuscript 'Patient information needs for transparent and trustworthy cardiovascular artificial intelligence: A qualitative study' has been provisionally accepted for publication in PLOS Digital Health.

Best regards,

Hadi Ghasemi

Academic Editor

PLOS Digital Health

**Additional Editor Comments (if provided):**

**Reviewer Comments (if any, and for reference):**

Reviewer's Responses to Questions

**Comments to the Author**

1. If the authors have adequately addressed your comments raised in a previous round of review and you feel that this manuscript is now acceptable for publication, you may indicate that here to bypass the “Comments to the Author” section, enter your conflict of interest statement in the “Confidential to Editor” section, and submit your "Accept" recommendation.

Reviewer #4: All comments have been addressed

2. Does this manuscript meet PLOS Digital Health’s publication criteria ? Is the manuscript technically sound, and do the data support the conclusions? The manuscript must describe methodologically and ethically rigorous research with conclusions that are appropriately drawn based on the data presented.

Reviewer #4: Yes

3. Has the statistical analysis been performed appropriately and rigorously?

Reviewer #4: N/A

4. Have the authors made all data underlying the findings in their manuscript fully available (please refer to the Data Availability Statement at the start of the manuscript PDF file)?

Reviewer #4: Yes

5. Is the manuscript presented in an intelligible fashion and written in standard English?

Reviewer #4: Yes

6. Review Comments to the Author

Reviewer #4: Well done for the significant efforts and clarification made

7. PLOS authors have the option to publish the peer review history of their article (what does this mean? ). If published, this will include your full peer review and any attached files.

**Do you want your identity to be public for this peer review?** For information about this choice, including consent withdrawal, please see our Privacy Policy .

Reviewer #4: No
